# Effects of CO$_2$ Treatments on Functional Carbon Efficiencies and Growth of Forest Tree Seedlings: A Study of Four Early-Successional Deciduous Species

**Axel Brisebois and John E. Major \***

Natural Resources Canada, Canadian Forest Service—Atlantic Forestry Centre, 1350 Regent St., Fredericton, NB E3B 5P7, Canada; axel.brisebois@nrcan-rncan.gc.ca
* Correspondence: john.major@nrcan-rncan.gc.ca

**Abstract:** Atmospheric CO$_2$ levels have been increasing, and these changes may result in differential adaptive responses in both genera and species and highlight the need to increase carbon sequestration. Ecophysiological and morphological responses of four early-successional deciduous species were examined under ambient CO$_2$ (aCO$_2$, 400 ppm) and elevated CO$_2$ (eCO$_2$, 800 ppm) treatments. The four species, all of which are used in restoration, were *Alnus viridis* subsp. *crispa* (Ait.) Turrill (green alder), *A. incana* subsp. *rugosa* (Du Roi) R.T. Clausen (speckled alder), *Betula populifolia* (Marshall) (gray birch), and *B. papyrifera* (Marshall) (white birch); all are from the same phylogenetic family, Betulaceae. We examined biochemical efficiencies, gas exchange, chlorophyll fluorescence, chlorophyll concentrations, foliar nitrogen (N), and growth traits. A general linear model, analysis of variance, was used to analyze the functional carbon efficiency and growth differences, if any, among genera, species, and provenances (only for growth traits). The alders had greater biochemical efficiency traits than birches, and alders upregulated these traits, whereas birches mostly downregulated these traits in response to eCO$_2$. In response to eCO$_2$, assimilation either remained the same or was upregulated for alders but downregulated for birches. Stomatal conductance was downregulated for all four species in response to eCO$_2$. Intrinsic water use efficiency was greater for alders than for birches. Alders exhibited a consistent upregulation of stem dry mass and height growth, whereas birches were somewhat lower in height and stem dry mass in response to eCO$_2$. Foliar N played an important role in relation to ecophysiological traits and had significant effects relative to genus (alders > birches) and CO$_2$ (aCO$_2$ > eCO$_2$), and a significant genus × CO$_2$ interaction, with alders downregulating foliar N less than did birches. Covariate analysis examining carbon efficiency traits in relation to foliar N showed clear functional responses. Both species in both genera were consistent in their ecophysiological and morphological responses to CO$_2$ treatments. There was supporting evidence that assimilation was sink-driven, which is related to a plant organ's ability to continue to grow and incorporate assimilates. The alders used in this study are actinorhizal, and the additional available foliar N, paired with increased stem dry mass sink activity, appeared to be driving upregulation of the carbon efficiencies and growth in response to eCO$_2$. Alders' greater carbon efficiencies and carbon sequestration in impoverished soils demonstrate that alders, as opposed to birches, should be used to accelerate ecological restoration in a world of increasing atmospheric CO$_2$.

**Keywords:** alders; birches; elevated CO$_2$; assimilation downregulation; foliar nitrogen

## 1. Introduction

Atmospheric CO$_2$ levels have been increasing, and these changes may result in differential adaptive responses in both genera and species. Quantifying and understanding plant ecophysiological and morphological responses to CO$_2$ will further our understanding of the various species useful for restoration, reforestation, and carbon sequestration. Growth and survival are determined by a wide range of physiological and morphological responses to

environmental factors, including $CO_2$ levels, soil moisture, and nutrient availabilities [1–3]. In addition, altered interspecific competitive relationships may impact plant community composition [4].

We examined four species from the same phylogenetic family, Betulaceae. Two were alders, namely, speckled alder (*Alnus incana* subsp. *rugosa* (Du Roi) R.T. Clausen) and green alder (*A. viridis* subsp. *crispa* (Ait.) Turrill). Both species are relatively short-lived, early-successional, deciduous shrubs native to north-eastern North America. Both species are shade intolerant, exhibit rapid early growth, and coppice following harvesting [5]. Both species are actinorhizal and form symbioses with *Frankia alni* bacteria, which can fix atmospheric N [6]. Speckled alder grows in wetter areas [7,8], and green alder in drier uplands, and both species grow on poor-quality soils [7]. We also examined two birches, white birch (*Betula papyrifera* (Marshall)) and gray birch (*B. populifolia* (Marshall)), both of which are fast-growing, shade intolerant, early successional deciduous species [9,10]. Gray birch's natural range is in eastern North America, where it is relatively short-lived, with a longevity of approximately 50 years. It occurs mostly on sandy or gravelly soils, whereas white birch is transcontinental, can live up to 120 years, [5] and occurs in a wide range of soil types.

Elevated $CO_2$ (e$CO_2$) often increases assimilation and water use efficiency in the short term, but by variable amounts, with the time depending on species, nutrient, and water availability [3,11,12]. A study investigating water relations in gray birch seedlings under e$CO_2$ (700 ppm) and drought conditions indicated increased tissue elasticity, which allowed plants to maintain positive turgor pressure at lower water potentials [13]. In an experiment testing leaf water relations of European black alder (*Alnus glutinosa* (L.) Gaertn.), black alder was found to have high leaf conductance and thus high leaf transpiration on both dry and wet sites, despite differing soil conditions [14]. This physiological trait was interpreted as being a strategy to maximize productivity where water was not typically limited, but it was noted that it may limit alder growth where groundwater cannot be exploited [14]. Although little genetic variation information exists in the literature on the four species selected for our study, low genetic differentiation among populations of *A. incana* (speckled alder) and *A. crispa* (green alder) in Quebec, Canada, has been suggested by [15]. This study also found large genetic differences between these species, emphasizing the possible effects of adaptation to differing ecological niches.

Our goal was to examine and compare growth and ecophysiological traits in four North American alder and birch species in response to $CO_2$ treatments. We examined biochemical efficiencies, gas exchange, chlorophyll fluorescence, chlorophyll concentration, foliar N, and growth (height growth of main stem and total stem dry mass) of the four species under ambient $CO_2$ (a$CO_2$, 400 ppm) and e$CO_2$ (800 ppm). We hypothesized that adaptive genetic differences would result in differences in functional carbon efficiencies in response to $CO_2$ treatments, and that these differences would occur at genus, species, and provenance levels (for height growth and total stem dry mass), and aimed to quantify the genetic variation of their responses. To test our hypothesis, we (1) quantified the functional carbon efficiencies and variation attributed to different taxonomic levels (genus, species, and provenance) for various ecophysiological and growth traits for green alder, speckled alder, gray birch, and white birch; (2) examined their ecophysiological and growth responses and possible interactions relative to the $CO_2$ treatments; and (3) used covariate analysis to examine ecophysiological traits in relation to foliar N and tested both $CO_2$ treatments and genera effects within these relationships.

## 2. Materials and Methods

### 2.1. Plant Material, Growing Conditions, and Treatment Delivery

Three to five seed sources (provenances) for each of the four hardwood species, namely, green alder, speckled alder, white birch, and gray birch, were used in our study (Table 1). Seed lots were taken from natural populations across three provinces of Canada, mostly from New Brunswick, with some drawn from Nova Scotia and Prince Edward Island.

Two to eight seeds were sown in late July of 2020, numbers in accordance to previous germination tests, and grown in hard-sided multi-pot trays of 90 cm$^3$ cell volume at the Atlantic Forestry Center (AFC) in Fredericton, New Brunswick (NB), Canada (45°52′ N, 66°31′ W). Early germinants were culled or moved to empty cells to ensure there was only one seedling per cell. Prior to frozen storage and well into being dormant, the hardwood stems were trimmed to approximately 10 cm, as had become the practice in our restoration experiments, in order to reduce leaf transpiration compared to root water absorption [16]. Seedlings were removed from trays and placed in bags, which were then boxed for cold storage at −4 to −5 °C in a freezer at AFC.

**Table 1.** Seed source and replicates of species and provenances of Betula and Alnus used in the 2021 CO$_2$ experiment. An 'X' denotes measurements taken on each seed source of each species.

| Species | Provenances | | # Per Chamber | Latitude | Longitude | Treatment Type | |
|---|---|---|---|---|---|---|---|
| | | | | | | Dry Mass, Foliar Nitrogen, and Foliar Pigments | A/Ci and Gas Exchange Parameters |
| *Betula populifolia* | Afton Road, PEI, CA | | 4 | 46.383 | −62.933 | X | X |
| | Bishop Mountain, NS, CA | | 4 | 45.033 | −64.983 | X | X |
| | Coles Island, NB, CA | | 4 | 45.900 | −65.717 | X | |
| | Mount Albian, PEI, CA | | 4 | 46.233 | −62.950 | X | |
| | Newmarket, NB, CA | | 4 | 45.805 | −66.956 | X | |
| *Betula papyrifera* | Fredericton, NB, CA | | 4 | 45.962 | −66.627 | X | X |
| | Lincoln, NB, CA | | 4 | 45.833 | −66.600 | X | |
| | Mount Pleasant, NB, CA | | 4 | 45.417 | −66.833 | X | |
| | Oromocto Lake, NB, CA | | 4 | 45.700 | −66.650 | X | X |
| | Wayerton, NB, CA | | 4 | 47.217 | −65.933 | X | |
| *Alnus viridis subsp. crispa* | Dipper Harbour West, NB, CA | | 4 | 45.100 | −66.433 | X | |
| | Jouriman Island, NB, CA | | 4 | 46.150 | −63.833 | X | |
| | Indian Falls Depot, NB, CA | | 4 | 47.383 | −66.333 | X | |
| | Lower Prince William, NB, CA | | 4 | 45.867 | −67.000 | X | X |
| | West Quaco, NB, CA | | 4 | 45.333 | −65.533 | X | X |
| *Alnus incana subsp. incana* | McDougall Lake, NB, CA | | 4 | 45.333 | −66.733 | X | X |
| | Shediac, NB, CA | | 4 | 46.233 | −64.600 | X | X |
| | Bulk | Millvale, PE, CA | | 46.400 | −63.400 | | |
| | | Valleyfield, PE, CA | 12 | 46.130 | −62.720 | X | |
| | | View Lake, NS, CA | | 44.530 | −65.350 | | |

Note that foliar chlorophyll pigment and foliar N were only measured on 2 replicates per chamber. For the 'bulk' provenance of *Alnus incana*, was made up of 3 seed sources as germination was low for the three.

To prepare the seedlings for the greenhouse experiment, on 18 June, the seedlings were removed from the freezer and stored in a 4 °C refrigerator to properly thaw the plants prior to their planting on 21 June 2021. Seedlings were randomly planted and grown individually in fabric root-control bags that measured 30 cm in diameter and 23 cm in height (Smart Pot PRO 5 Gallon, High Caliper Growing Systems, Oklahoma City, OK, USA) and were filled with screened sand (M.W. Price & Sons, Charter's Settlement, NB, CA, USA). Bulk soil samples were taken on 31 August 2022, and sent for nutrient and soil texture analysis at the University of New Brunswick in Fredericton. The soil texture properties were, on average, 1.8% clay, 7.8% silt, and 90.4% sand, with no significant nutrient or pH differences between blocks (Table 2). The root-control bags were arranged on a sand-covered gravel surface inside eight custom growth chambers, each measuring 2.3 m wide, 4.25 m long, and 2.1 m in height [3]. Chambers were covered in UVA-clear 4 mil polyethylene film. Air inside the chambers was delivered and circulated via positive pressure fans, with air conditioner units placed outside each chamber. Plants within the chambers were grown under natural light, with light levels inside the chambers measuring approximately 65% of the ambient levels determined by methods presented by [17] ("A

simple and efficient method to estimate microsite light availability under a forest canopy"). Watering was regulated using a dripper system of individual spigots in each root bag, which were connected through a common hose system. Sensors within each chamber were connected to a custom-made logging and control system which continuously monitored the sensors within the chambers, including measurements of air temperature and humidity (model HMP 155, Vaisala, Vantaa, Finland, and Model AM2302, Adafruit Industries, New York, NY, USA); soil moisture and temperature (5TM, Decagon Devices, Pullman, WA, USA); and $CO_2$ concentration (Model 840A, Li-COR Inc, Lincoln, NE, USA), which was measured mid-chamber.

**Table 2.** Soil nutrient and texture properties (mean $\pm$ SE, $n = 16$).

| Organic Matter (%) | Carbon (%) | Nitrogen (%) | Phosphorus (ppm) | Potassium (meq/100 g) | Calcium (meq/100 g) | Magnesium (meq/100 g) |
|---|---|---|---|---|---|---|
| $0.568 \pm 0.049$ | $2.775 \pm 0.213$ | $0.119 \pm 0.002$ | $6.375 \pm 1.002$ | $0.049 \pm 0.007$ | $0.568 \pm 0.049$ | $0.104 \pm 0.01$ |
| **pH** | **C:N Ratio** | **Sand (%)** | **Silt (%)** | **Clay (%)** | | |
| $6.045 \pm 0.079$ | $23.389 \pm 1.658$ | $90.4 \pm 3.09$ | $7.8 \pm 2.91$ | $1.8 \pm 0.75$ | | |

In each chamber, four replicates of each provenance for each species (Table 1) were distributed randomly. Two $CO_2$ treatments (ambient 400 ppm and elevated 800 ppm) were randomly assigned to four growth chambers, along within a replicate block of each treatment type. The two $CO_2$ treatments comprised ambient treatments (a$CO_2$—No $CO_2$ added, ~400 ppm) and elevated treatments (e$CO_2$—$CO_2$ regulated at ~800 ppm). Elevated-$CO_2$ chambers were regulated through the opening or closing of solenoid valves to control the level of $CO_2$ being delivered via the air stream entering the chamber. Irrigation was maintained at ~15–20% VWC. Irrigation was manually controlled through valves used to deliver water as needed to maintain the desired levels. $CO_2$ treatments began on 21 June 2021, directly after seedlings were planted. Starter fertilizer (Plant-Prod "Forestry Starter" 11:41:8, 287 g/25 L, +250 mL of MgNiFeCa) was supplied to the plants, at half-strength, to all treatments in equal proportions, via the dripper system (30 min = 1 L) on 23 June (15 min), 24th (15 min), 30th (15 min) and 5 July (20 min), 2021. Grower fertilizer (Plant-Prod "Forestry Special" 20:8:20, 625 g/50 L, +1250 mL of MgNiFeCa) was supplied to the plants, at half-strength, to all treatments in equal proportions, via the dripper system (30 min = 1 L) on 9 July (15 min), 6 August (10 min), 27 August (5 min) and 11 October (5 min), 2021.

*2.2. Biochemical Efficiencies, Fluorescence, and Gas-Exchange Traits*

An LI-6800 portable gas-exchange and fluorescence system (LiCOR, Lincoln, NE, USA) was used to measure assimilation ($A$) to internal $CO_2$ ($A/C_i$) in situ, in order to develop curves to determine biochemical efficiencies using graphical presentation and curve-fitting algorithms; their interpretations are described later. Measurements were randomly made on four species from three provenances, using two blocks and two replicates within a block, for two $CO_2$ treatments, between 29 September and 19 October (days 101–121 of treatment), resulting in ($4 \times 3 \times 2 \times 2 \times 2$) 96 $A/C_i$ curves. The instrument was programmed to maintain leaf temperature at 18–22 °C, with an air flow of 600 $\mu$mol$\cdot$ s$^{-1}$, and a saturating photosynthetic photon flux density (PPFD) level of 1000 umol$\cdot$m$^{-2}\cdot$s$^{-1}$. The relative humidity averaged 58%, but was adjusted situationally to maintain a vapor pressure deficit (VPD) of approximately 1.75 kPa in the leaf chamber. Gas exchange (*GE*) measurements for A/Ci curves were made at 11 different $CO_2$ levels, which were, in order, 400, 330, 260, 190, 120 50, 400, 400, 600, 800, 1200, 1700, and 2200 ppm. Note, 400 ppm was measured three times at the beginning and twice after 50 ppm to ensure the plant had acclimated to such a change. When each leaf was initially placed in the chamber, an ~15 min period was used to stabilize chamber conditions prior to leaf measurements. A settling time of at least 2 min occurred once each $CO_2$ level was reached before measurements were recorded. The $A/C_i$ response curves were fitted using a commercially available Excel-based calculator

called "*A/Ci* curve fitting utility", version 2007.1, developed by the authors of [18]. The program uses algorithms to fit estimations for the maximum rate of carboxylation ($V_{cmax}$), maximum rate of photosynthetic electron transport ($J_{max}$), and triose phosphate use (*TPU*). To generate estimations and fit the model, the program utilizes a least-squares fit using the solver add-in by Microsoft Excel.

Additionally, gas exchange (*GE*) and fluorescence parameters were compared at a common and intermediate $CO_2$ level of 600 ppm, which is the mid-point between $CO_2$ treatments, to examine physiological downregulation. *A* at 600 ppm $CO_2$ is thus abbreviated as $A_{600}$. Other GE parameters examined include stomatal conductance at 600 ppm $CO_2$ ($G_{wv600}$), and intrinsic water use efficiency at 600 ppm $CO_2$ ($iWUE_{600}$); the latter of which was calculated as $A_{600}$ divided by $G_{wv600}$. Fluorescence data was also collected from measurements obtained at 600 ppm. Quantum yield of photosystem 2 ($\Phi_{PSII}$) was calculated using light-adapted maximum fluorescence ($Fm'$) and steady state fluorescence ($Ft$). Thus, $\Phi_{PSII} = (Fm' - Ft)/Fm'$. Relative fluorescence (*RelFlu*) used light-adapted minimum fluorescence ($Fo'$); thus, $RelFlu = (Ft - Fo')/Fm'$.

### 2.3. Leaf Chlorophyll, Leaf Nitrogen, and Growth Assessment

Leaf chlorophyll and N samples were taken in September 2021 by harvesting two leaves from one individual of each provenance in each chamber. The leaves harvested were mature and taken from the top one-third of each plant. These two leaves were placed into sealed, labeled plastic bags. Once this was completed, all bags were placed into a $-80$ °C freezer for preservation for later chlorophyll and foliar N analysis. Chlorophyll extractions were randomly made on four species with five provenances (three for speckled alder), based on two blocks and two replicates, given two $CO_2$ treatments ($n = 144$), which were performed 8 February–14 February. Leaf samples were removed from the $-80$ °C freezer and kept in a cooler filled with ice to maintain darkness and preserve leaf tissue. Small samples were cut off both leaves mid-blade, as taken from each bag, weighed to the nearest 0.00001 g (AT261 DeltaRange, Mettler Toledo Inc., Columbus, OH, USA), and then placed in a micro centrifuge tube. Note that chlorophyll pigment and *N* concentrations were the means of the two subsamples taken from each plant. Additionally, empty tubes were included as blanks.

To extract the chlorophyll, 1.5 mL of dimethyl-formamide (DMF) was added to each leaf sample, in addition to the blank tubes, to act as a control when scanned for absorbance. The tubes were then sealed, covered in aluminum foil to prevent light degradation, and placed on an orbital shaker set to 125 rpm (Innova 2000 platform shaker, New Brunswick Scientific, Edison, NJ, USA) for 23 h at room temperature to ensure proper mixing and extraction of chlorophyll. The microplate used for absorbance (Corning COSTAR 3364 Storage Plate, ETC) was first scanned using a spectrophotometer (Ultrospec 2100 pro, Biochrom Ltd., Holliston, MA, USA) at 480, 647, and 664 nm wavelengths for the later correction of absorbance readings. Next, each sample was inverted to ensure proper mixing before 40 µL of liquid was extracted from each tube and added to the same microplate, which was scanned and diluted with 80 µL of DMF. Absorbance readings were again obtained at 480, 647, and 664 nm wavelengths. All blank readings were <0.003 for each wavelength after the plate correction was made. Concentrations ($ug \cdot mL^{-1}$) of chlorophyll *a*, chlorophyll *b*, total chlorophyll (*TCC*, chlorophyll *a* + *b*), and total carotenoids (*CAR*) were calculated using the previously weighed fresh weight ($ug \cdot g^{-1}$) of each sample.

Total foliar N was determined for each remaining leaf sample using an elemental analyzer (CNS-2000, LECO Corporation, St. Joseph, MI, USA) service provided by the Laboratory for Forest Soils and Environmental Quality at the University of New Brunswick.

Above-ground biomass was harvested on 16 November 2021 by cutting each plant at 2 cm above soil level. Heights were taken for each plant's main stem by measuring with a metric tape, as measured to the nearest 0.1 cm, from soil level to the top of the main stem. The initial height of the planted seedling (as previously stated, ~10 cm) was then subtracted from the measured height to account for any deviation in trimmed seedling

height. The stems and branches were then stripped of any remaining leaves (which were then discarded), and the stems were placed in labeled paper bags to be dried in ovens at 65 °C for a minimum of 72 h before being weighed using a precision scale capable of measuring to within 0.01 g (accu-4102, Fisher Scientific, Waltham, MA, USA).

*2.4. Statistical Analyses*

This experiment utilized a randomized block design. The fixed effects in the physiological model were genus, species, and $CO_2$ treatments. Growth chamber (block) and replicate seedlings were random effects. We utilized a general linear model (GLM) to conduct mixed-effects analyses of variance (ANOVA) with nested factors, using Systat version No.13.00.05 (San Jose, CA, USA). The first model was used for biochemical efficiency parameters, gas exchange parameters, chlorophyll fluorescence, chlorophyll pigments, and N concentration:

$$Y_{ijkmn} = \mu + B_i + G_j + C_k + GC_{jk} + S_{m(j)} + SC_{m(j)k} + e_{ijkmn} \qquad (1)$$

where $Y_{ijkmn}$ denotes the dependent variable of seedlings of the $i^{th}$ growth chamber (block), of the $j^{th}$ genus, of the $k^{th}$ $CO_2$ treatment, of the $m^{th}$ species, and the $n^{th}$ seedling, with $\mu$ being the overall mean. $B_i$ refers to the effect of the $i^{th}$ growth chamber (i = 1, 2), $G_j$ is the effect of the $j^{th}$ genus (j = 1, 2), and $C_k$ is the effect of the $k^{th}$ $CO_2$ treatment (k = 1, 2). $GC_{jk}$ is the interaction effect between genus *j* and $CO_2$ treatment *k*. $S_{m(j)}$ is the effect of species *m* (m = 1, 2) nested in genus *j*. $SC_{m(j)k}$ is the interactive effect of species *m* nested in genus *j* with $CO_2$ treatment *k*. Lastly, $e_{ijkmn}$ is the random error component, incorporating interactions with the growth chamber factor and the variation among seedlings.

For growth traits, the following model was used; the only additional factor was provenance, which was considered a fixed effect, while all others remained the same:

$$Y_{ijkmno} = \mu + B_i + G_j + C_k + GC_{jk} + S_{m(j)} + SC_{m(j)k} + P_{o(m)} + PC_{o(m)k} + e_{ijkmno} \qquad (2)$$

$Y_{ijklmn}$ denotes the dependent seedling of the $i^{th}$ growth chamber; of the $j^{th}$ genus; of the $k^{th}$ $CO_2$ treatment; of the $m^{th}$ species, or $o^{th}$ provenance; and the $n^{th}$ seedling; with $\mu$ being the overall mean. $B_i$ refers to the effect of the $i^{th}$ growth chamber (*i* = 1, 2), $G_j$ is the effect of the $j^{th}$ genus (*j* = 1, 2), and $C_k$ is the effect of the $k^{th}$ $CO_2$ treatment (*k* = 1, 2). $GC_{jk}$ is the interactive effect between genus *j* and $CO_2$ treatment *k*. $S_{m(j)}$ is the effect of species *m* (*m* = 1…4) nested in genus *j*. $SC_{m(j)k}$ is the interactive effect of species *m* nested in genus *j* with $CO_2$ treatment *k*. $P_{o(m)}$ is the effect of provenance *o* (*o* = 1…3 or 1…5 depending on species) nested in species *m*. $PC_{o(m)k}$ is the interactive effect of provenance *o* nested in species *m* with $CO_2$ treatment *k*. Lastly, $e_{ijkmno}$ is the random error component, incorporating interactions with the growth chamber factor and the variation among seedlings.

Effects were considered statistically significant at the *p* = 0.05 level, although all *p*-values are listed for the reader's interpretation. Variance component analysis was conducted for the ANOVA tables using the sums of squares following methods outlined in "Variance Component Analysis" in [19]. The statistical assumptions of residual normality and equal variance were satisfied prior to running either model. The general linear model from Systat version No.13.00.05 (San Jose, CA, USA) was used for these analyses, and if the source of variation for species was significant (*p* = 0.050), the Tukey mean separation test was used for post hoc analysis.

Covariate analysis was used to evaluate the relationships among species' mean physiological traits and to test $CO_2$ treatment and genus effects. In these analyses, the dependent variable (e.g., $A_{600}$) was examined in relation to three sources of variation: (1) covariate (e.g., foliar N), (2) independent effect (e.g., $CO_2$ treatment), and (3) independent

effect × covariate (e.g., $CO_2$ treatment × foliar N). The analyses were performed based on the following model:

$$Y_{ij} = B_0 + B_{0i} + B_1 X_{ij} + B_{1i} X_{ij} + e_{ij} \qquad (3)$$

where $Y_{ij}$ is the dependent trait of the $i^{th}$ species of the $j^{th}$ genus treatment. $B_0$ and $B_1$ are average regression coefficients, $B_{0i}$ and $B_{1i}$ the treatment-specific coefficients, $X_{ij}$ the independent variable (e.g., $CO_2$ treatment), and $e_{ij}$ the error term. Results were still considered statistically significant at $p = 0.050$, although individual $p$-values are provided for all traits so that readers can make their own interpretations of significance, particularly if these results were between $p = 0.050$ and $p = 0.100$, which we would consider marginally significant. Simple linear regression was used to examine change in assimilation to change in total stem dry mass from $aCO_2$ to $eCO_2$.

## 3. Results

### 3.1. Biochemical Efficiencies and Gas Exchange Parameters

For $V_{cmax}$, genus and species were significant, accounting for 32.4 and 10.0% of the total variation, respectively (Table 3). Alders had significantly greater $V_{cmax}$ than birches, with 29.8 and 21.3 µmol·m$^{-2}$·s$^{-1}$, respectively, a 40% difference (Figure 1A). For green alder, speckled alder, gray birch, and white birch, $V_{cmax}$ values were 28.3, 31.4, 24.3, and 18.4 µmol·m$^{-2}$·s$^{-1}$, respectively. In response to $eCO_2$, alders upregulated $V_{cmax}$ from 28.6 to 31.1 (8.7%) and birches downregulated $V_{cmax}$ from 22.3 to 20.3 (−9.0%), a near-significant genus × $CO_2$ interaction ($p = 0.066$). For $J_{max}$, genus, species, and genus × $CO_2$ interaction were significant, accounting for 40.2, 5.3, and 4.1% of total variance, respectively (Table 3). The significant genus × $CO_2$ interaction was due to a rank change in genus response to $eCO_2$. Alders upregulated $J_{max}$ from 62.7 to 65.4 (4.3%), and birches downregulated from 48.9 to 38.7 (−20.9%). Despite this, alders had significantly greater $J_{max}$ than birches, with 64.0 and 43.8 µmol·m$^{-2}$·s$^{-1}$, respectively, a difference of 46% (Figure 1B). For green alder, speckled alder, gray birch, and white birch, the $J_{max}$ values were 61.6, 66.5, 48.4, and 39.2 µmol·m$^{-2}$·s$^{-1}$, respectively. For *TPU*, genus, species, and $CO_2$ × genus were significant, accounting for 39.0, 6.4, and 4.4% of total variance, respectively (Table 3). Like $J_{max}$, the significant genus × $CO_2$ interaction was due to alders upregulating and birches downregulating in response to $eCO_2$. Alders had greater *TPU* than birches, with 4.8 and 3.2 µmol·m$^{-2}$·s$^{-1}$, respectively, a 50% difference (Figure 1C). For green alder, speckled alder, gray birch, and white birch, *TPU* values were 4.6, 5.0, 3.6, and 2.8 µmol·m$^{-2}$·s$^{-1}$, respectively.

**Table 3.** Biochemical efficiency trait ANOVAs, including source of variation, degrees of freedom (df), mean square values (MS), variance components (VC), $p$-values, and coefficient of determination ($R^2$). All $p$-values < 0.05 are in bold print.

| Source of Variation | df | * $V_{cmax}$ (µmol m$^{-2}$s$^{-1}$) | | | * $J_{max}$ (µmol m$^{-2}$s$^{-1}$) | | | * *TPU* (µmol m$^{-2}$s$^{-1}$) | | |
|---|---|---|---|---|---|---|---|---|---|---|
| | | MS | VC (%) | *p*-Value | MS | VC (%) | *p*-Value | MS | VC (%) | *p*-Value |
| Block | 1 | 43.4 | 0.9 | 0.255 | 192.8 | 0.8 | 0.234 | 0.7 | 0.5 | 0.361 |
| Genus | 1 | 1645.8 | 32.4 | **<0.001** | 9270.5 | 40.2 | **<0.001** | 53.8 | 39.0 | **<0.001** |
| $CO_2$ | 1 | 1.5 | 0.0 | 0.831 | 312.9 | 1.4 | 0.131 | 0.4 | 0.3 | 0.463 |
| $CO_2$ × Genus | 1 | 114.4 | 2.3 | 0.066 | 952.6 | 4.1 | **0.009** | 6.1 | 4.4 | **0.008** |
| Species (Genus) | 2 | 252.6 | 10.0 | **0.001** | 607.5 | 5.3 | **0.014** | 4.4 | 6.4 | **0.006** |
| $CO_2$ × Species (Genus) | 2 | 30.6 | 1.2 | 0.399 | 66.5 | 0.6 | 0.611 | 0.5 | 0.8 | 0.523 |
| Error | 82 | 32.9 | 53.2 | | 134.2 | 47.7 | | 0.8 | 48.5 | |
| $R^2$ | | | 0.474 | | | 0.529 | | | 0.522 | |

* Abbreviations—Maximum rate of carboxylation ($V_{cmax}$), maximum rate of electron transport ($J_{max}$), triose phosphate utilization (*TPU*).

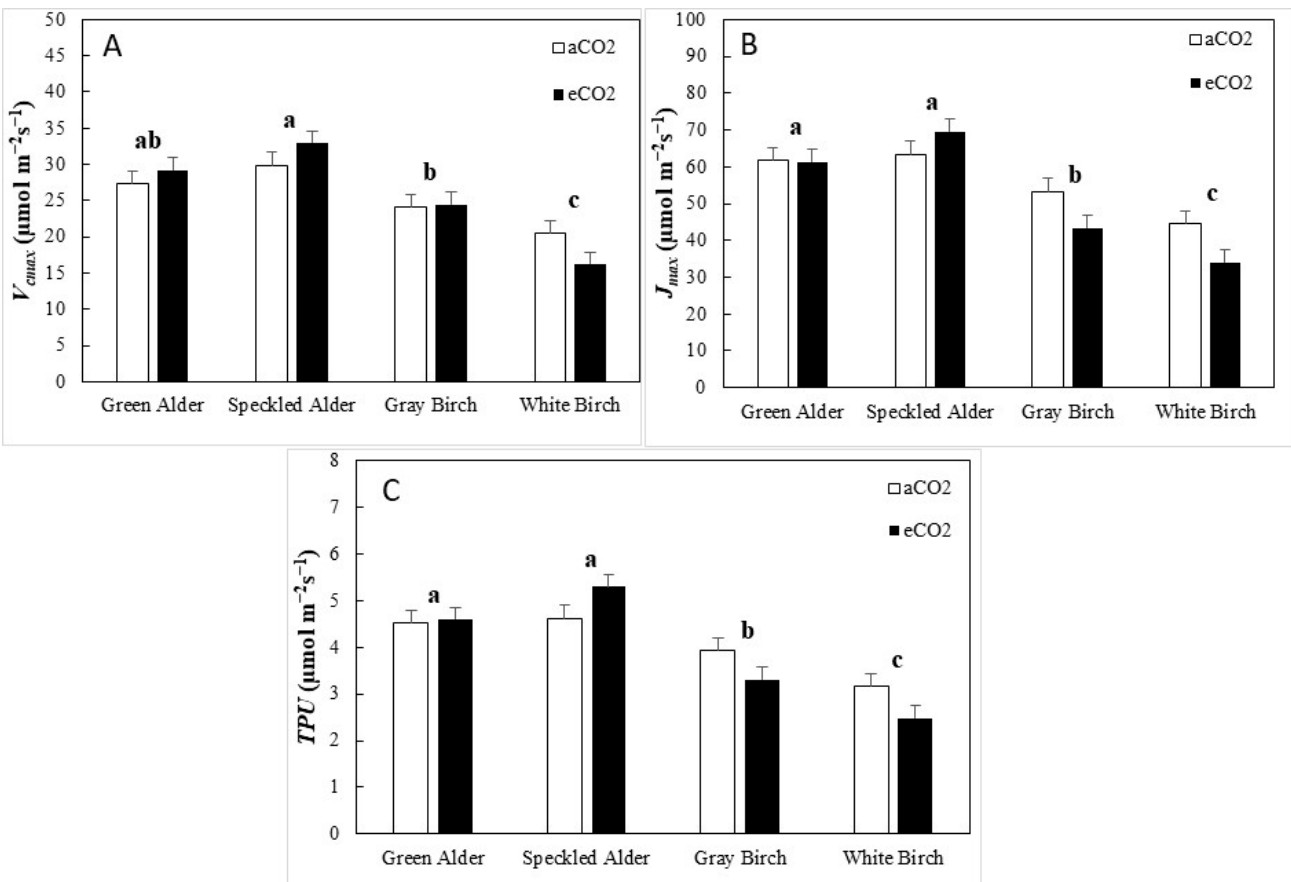

**Figure 1.** Biochemical efficiency traits (mean $\pm$ *SE*, *n* = 96), including, (**A**)maximum rate of carboxylation ($V_{cmax}$), (**B**) maximum rate of electron transport ($J_{max}$), and (**C**) triose phosphate utilization (*TPU*), by species and $CO_2$ treatment type (a$CO_2$ = ambient $CO_2$, e$CO_2$ = elevated $CO_2$). A post hoc Tukey's mean separation test was performed on species (*p* = 0.050) for each analysis where species was significant (Table 3). Species with the same letter are not significantly different.

For $A_{600}$, variations in $CO_2$, genus, species, and genus $\times$ $CO_2$ interaction were significant, accounting for 3.5, 41.2, 3.4 and 5.9% of total variation, respectively (Table 4). The interaction between genus and $CO_2$ was significant because alders remained statistically the same under e$CO_2$ (+3%), whereas birches downregulated from 10.3 to 7.3 $\mu mol \cdot m^{-2} \cdot s^{-1}$ (−29%). Alders had significantly greater $A_{600}$ than birches at 13.3 and 8.8 $\mu mol \cdot m^{-2} \cdot s^{-1}$, respectively, a 51% difference (Figure 2A). For green alder, speckled alder, gray birch, and white birch, $A_{600}$ values were 12.8, 13.7, 9.6, and 8.0 $\mu mol \cdot m^{-2} \cdot s^{-1}$, respectively. For $G_{wv600}$, only $CO_2$ was significant (Table 4). This was due to all species downregulating $G_{wv600}$, which decreased on average by 0.096 $\mu mol \cdot m^{-2} \cdot s^{-1}$ (−22.9%) under e$CO_2$ (Figure 2B). The $G_{wv600}$ for green alder, speckled alder, gray birch, and white birch were 0.347, 0.365, 0.354, and 0.379 $\mu mol \cdot m^{-2} \cdot s^{-1}$, respectively. For $iWUE_{600}$, only genus was significant (Table 4). Alders had a significantly greater $iWUE_{600}$, with an average of 41.6, whereas birches were 25.8, a 41% difference (Figure 2C). The $iWUE_{600}$ for green alder, speckled alder, gray birch, and white birch were 41.5. 41.7, 29.9, and 21.6, respectively.

**Table 4.** Gas-exchange trait ANOVAs including source of variation, degrees of freedom (df), mean square values (MS), variance components (VC), *p*-values, and coefficient of determination ($R^2$). All *p*-values <0.05 are in bold print.

| Source of Variation | df | $* A_{600}$ (μmol m$^{-2}$s$^{-1}$) | | | $* G_{wv600}$ (μmol m$^{-2}$s$^{-1}$) | | | $* iWUE_{600}$ | | |
|---|---|---|---|---|---|---|---|---|---|---|
| | | MS | VC (%) | *p*-Value | MS | VC (%) | *p*-Value | MS | VC (%) | *p*-Value |
| Block | 1 | 2.6 | 0.2 | 0.519 | 0.015 | 1.1 | 0.287 | 204.3 | 0.9 | 0.317 |
| Genus | 1 | 453.6 | 41.2 | **<0.001** | 0.002 | 0.1 | 0.677 | 5659.4 | 23.9 | **<0.001** |
| CO$_2$ | 1 | 38.4 | 3.5 | **0.014** | 0.208 | 15.4 | **<0.001** | 52.5 | 0.2 | 0.611 |
| CO$_2$ × Genus | 1 | 64.4 | 5.9 | **0.002** | 0.028 | 2.1 | 0.148 | 223.7 | 0.9 | 0.295 |
| Species (Genus) | 2 | 18.7 | 3.4 | 0.051 | 0.005 | 0.8 | 0.662 | 386.9 | 3.3 | 0.153 |
| CO$_2$ × Species (Genus) | 2 | 2.1 | 0.4 | 0.705 | 0.004 | 0.6 | 0.741 | 122.9 | 1 | 0.545 |
| Error | 82 | 6.1 | 45.4 | | 0.013 | 79.9 | | 201.2 | 69.7 | |
| $R^2$ | | | 0.55 | | | 0.199 | | | 0.311 | |

* Abbreviations—Assimilation at 600 ppm ($A_{600}$), stomatal conductance at 600 ppm ($G_{wv600}$), and intrinsic water-use efficiency at 600 ppm ($iWUE_{600}$).

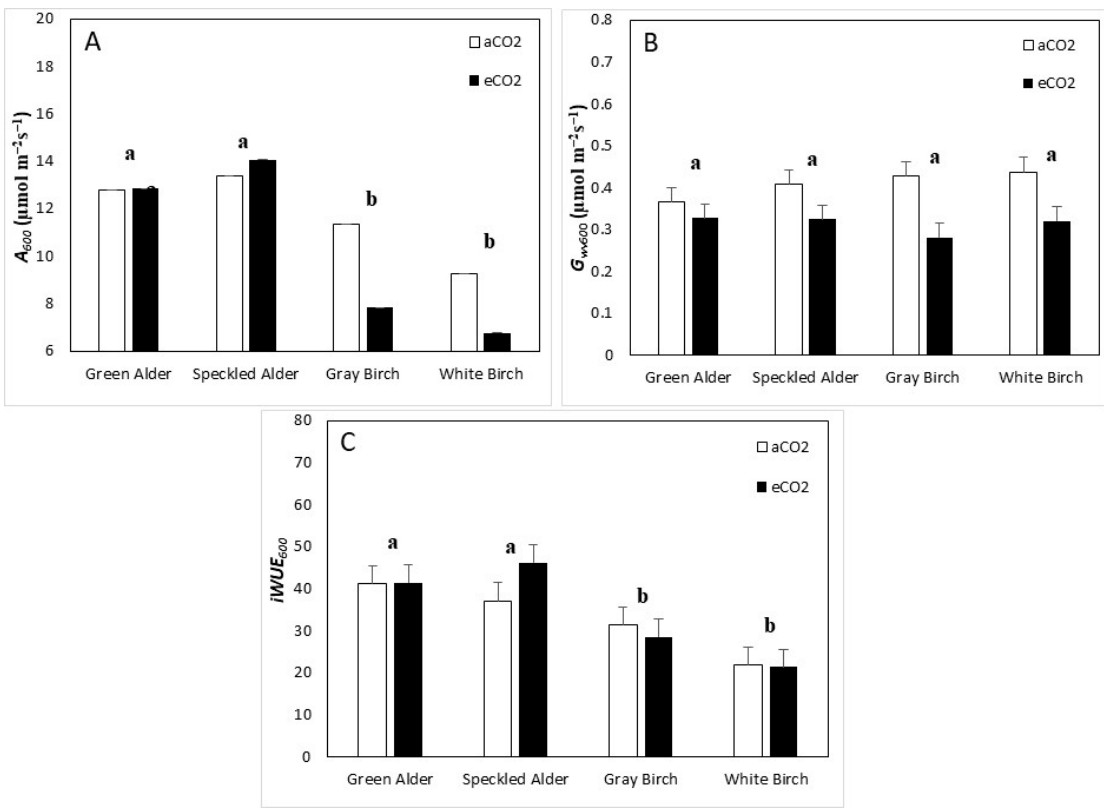

**Figure 2.** Gas-exchange traits (mean ± *SE*, *n* = 96), including, (**A**) assimilation at 600 ppm ($A_{600}$), (**B**) stomatal conductance at 600 ppm ($G_{wv600}$), and (**C**) intrinsic water-use efficiency at 600 ppm ($iWUE_{600}$), by species and CO$_2$ treatment type (aCO$_2$ = ambient CO$_2$, eCO$_2$ = elevated CO$_2$). A post hoc Tukey's mean separation test was performed on species (*p* = 0.05) for each analysis where species was significant (Table 4). Species with the same letter are not significantly different.

For quantum yield of PSII ($Φ_{PSII}$), variations in genus, species, CO$_2$, and genus and CO$_2$ interaction were significant, accounting for 42.9, 4.1, 2.4, and 4.8% of the total variation, respectively (Table 5). Alders upregulated/remained the same from aCO$_2$ to eCO$_2$ from 0.34 to 0.35 (+3%), whereas birches downregulated from 0.26 to 0.19 (−26%) (Figure 3A). Alders had a significantly greater $Φ_{PSII}$ than birches at 0.34, as compared to birches at 0.23,

a 51% difference. For lower steady-state fluorescence (*RelFlu*), variation in both genus and species were significant, accounting for 23.7 and 11.5% of the total variation, respectively (Table 5). Birches had greater average *RelFlu*, at 0.382, compared to alders at 0.319, a 20% difference. As for green alder, speckled alder, gray birch, and white birch, they had *RelFlu* values of 0.34, 0.30, 0.36, and 0.41, respectively (Figure 3B).

**Table 5.** Fluorescence trait ANOVAs, including source of variation, degrees of freedom (df), mean square values (MS), variance components (VC), *p*-values, and coefficient of determination ($R^2$). All *p*-values <0.05 are in bold print.

| Source of Variation | df | * $\Phi_{PSII}$ | | | * *RelFlu* | | |
|---|---|---|---|---|---|---|---|
| | | MS | VC (%) | *p*-Value | MS | VC (%) | *p*-Value |
| Block | 1 | 0.001 | 0.1 | 0.709 | <0.001 | <0.1 | 0.894 |
| Genus | 1 | 0.303 | 42.9 | **<0.001** | 0.091 | 23.7 | **<0.001** |
| $CO_2$ | 1 | 0.017 | 2.4 | **0.039** | <0.001 | <0.1 | 0.835 |
| $CO_2 \times$ Genus | 1 | 0.034 | 4.8 | **0.004** | 0.005 | 1.3 | 0.206 |
| Species (Genus) | 2 | 0.015 | 4.1 | **0.027** | 0.022 | 11.5 | **0.001** |
| $CO_2 \times$ Species (Genus) | 2 | 0.001 | 0.4 | 0.714 | 0.001 | 0.8 | 0.612 |
| Error | 82 | 0.004 | 45.2 | | 0.003 | 62.8 | |
| $R^2$ | | | 0.552 | | | 0.373 | |

\* Abbreviations—Quantum yield of photosystem II ($\Phi PSII$) and steady-state fluorescence (*RelFlu*).

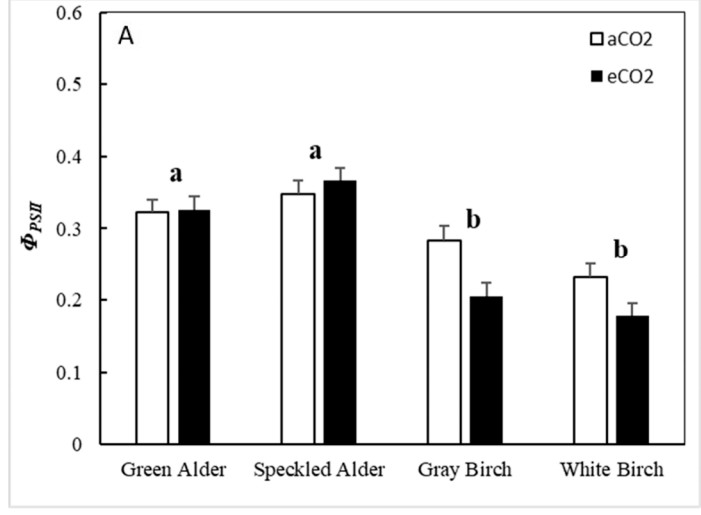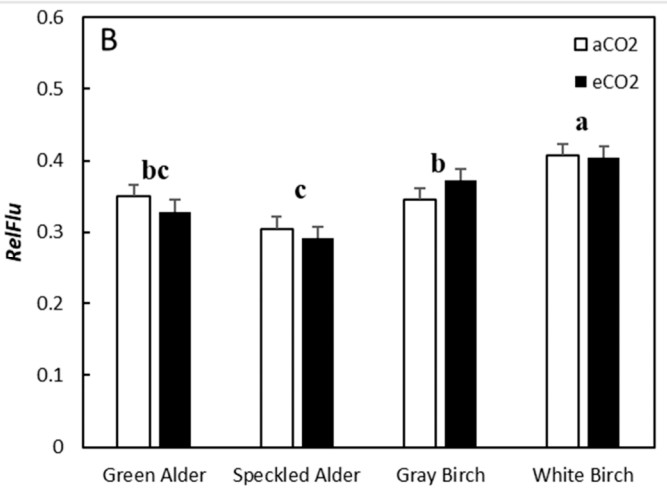

**Figure 3.** Fluorescence traits (mean $\pm$ *SE*, *n* = 96), including, (**A**) quantum yield of photosystem II ($\Phi PSII$), and (**B**) steady-state fluorescence *(RelFlu)*, by species and $CO_2$ treatment type (a$CO_2$ = ambient $CO_2$, e$CO_2$ = elevated $CO_2$). A post hoc Tukey's mean separation test was performed on species (*p* = 0.05) for each analysis where species was significant (Table 5). Species with the same letter are not significantly different.

### 3.2. Foliar Chlorophyll Traits and Foliar Nitrogen Analysis

The ANOVA results for *CHLa* and *CHLb* were the same as those for total chlorophyll concentration (*TCC*); thus, only *TCC* results are presented here. For *TCC*, variations in genus, species, and $CO_2$ were significant, accounting for 10.4, 7.3, and 5.5% of total variation, respectively (Table 6). Alders had a significantly greater *TCC*, at 3.1 mg/g, compared to birches at 2.4 mg/g, a 27.3% difference (Figure 4A). The average *TCC* of green alder, speckled alder, gray birch, and white birch were 2.9, 3.3, 2,8, and 2.1 mg/g, respectively. Total chlorophyll concentration downregulated in response to e$CO_2$ by 0.34 (−11%), on average.

**Table 6.** Foliar pigment and nitrogen concentration traits ANOVAs, including the source of variation, degrees of freedom (df), mean square values (MS), variance components (VC), *p*-values, and coefficient of determination ($R^2$). All *p*-values < 0.05 are in bold print.

| Source of Variation | df | Total Chlorophyll (mg/g) | | | Carotenoids (mg/g) | | | Nitrogen (%) | | |
|---|---|---|---|---|---|---|---|---|---|---|
| | | MS | VC (%) | *p*-Value | MS | VC (%) | *p*-Value | MS | VC (%) | *p*-Value |
| Block | 1 | 2.444 | 1.7 | 0.087 | 0.170 | 7.5 | **<0.001** | 1.273 | 2.3 | **0.005** |
| Genus | 1 | 14.156 | 9.6 | **<0.001** | 0.256 | 11.3 | **<0.001** | 17.245 | 31.1 | **<0.001** |
| $CO_2$ | 1 | 3.864 | 2.6 | **0.032** | 0.001 | <0.1 | 0.781 | 10.625 | 19.2 | **<0.001** |
| $CO_2$ × Genus | 1 | 2.384 | 1.6 | 0.091 | 0.067 | 3.0 | **0.019** | 2.583 | 4.7 | **<0.001** |
| Species (Genus) | 2 | 8.026 | 10.9 | **<0.001** | 0.094 | 8.3 | **0.001** | 0.924 | 3.3 | **0.004** |
| $CO_2$ × Species (Genus) | 2 | 0.565 | 0.8 | 0.505 | 0.01 | 0.9 | 0.417 | 0.621 | 2.2 | **0.022** |
| Error | 131 | 0.822 | 72.9 | | 0.012 | 68.9 | | 0.157 | 37.2 | |
| $R^2$ | | | 0.264 | | | 0.310 | | | 0.629 | |

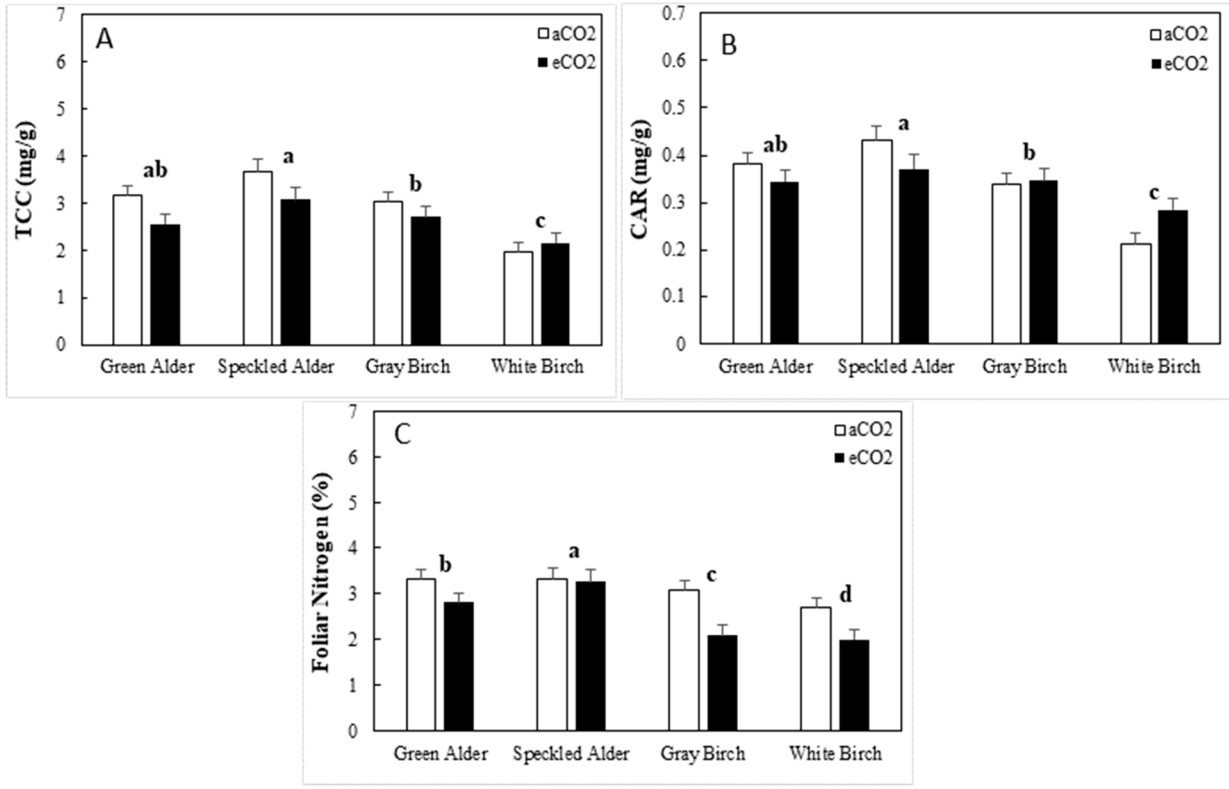

**Figure 4.** Foliar pigment and nitrogen concentration traits (mean ± *SE*, *n* = 144) including, (**A**) total foliar chlorophyll concentration (TCC), (**B**) foliar carotenoids (CAR), and (**C**) foliar nitrogen concentration by species and $CO_2$ treatment type (a$CO_2$ = ambient $CO_2$, e$CO_2$ = elevated $CO_2$). A post hoc Tukey's mean separation test was performed on species (*p* = 0.05) for each analysis where species was significant (Table 6). Species with the same letter are not significantly different.

For carotenoid concentration (*CAR*), genus, species, and the genus × $CO_2$ interaction were significant, accounting for 12.7, 6.3, and 3.0% of total variation, respectively (Table 6). The interaction was due to alders downregulating *CAR* by 0.06 mg/g (−14.8%), and birches remaining statistically the same (gray birch) or upregulating (white birch), on average, by 0.03 mg/g (+11.2%). Alders had significantly greater *CAR*, at 0.381, compared to birches at 0.292, a 30.5% difference. The *CAR* values of green alder, speckled alder, gray birch, and white birch were 0.36, 0.40, 0.34, and 0.25 mg/g, respectively (Figure 4B).

For foliar N, genus, species, $CO_2$, genus × $CO_2$ and species × $CO_2$ were significant, accounting for 31.1, 3.3, 19.2, 4.7, and 2.2% of total variation, respectively (Table 6). The genus × $CO_2$ interaction was a result of alders remaining statistically the same under $eCO_2$, whereas birches downregulated N from 2.9 to 2.1% (−28.3%). Alders had a significantly greater N than birches, at 3.2% compared to 2.5%, a 30.4% difference. The N values of green alder, speckled alder, gray birch, and white birch were 3.1, 3.3, 2.6, and 2.4%, respectively (Figure 4C). The species × $CO_2$ interaction was because green alder downregulated under $eCO_2$, from 3.2% to 2.8% (−16%), and speckled alder from 3.31% to 3.27 (−1%). Gray birch and white birch downregulated from 3.1 to 2.1% (−32%) and 2.7 to 2.0%, respectively (−26%).

### 3.3. Main Stem Height Growth and Total Stem Dry Mass

For main stem height growth, species and genus × $CO_2$ interaction were significant, accounting for 25.3 and 11.7% of total variation, respectively (Table 7). The $CO_2$ × genus interaction was a result of alders' upregulated height growth under $eCO_2$; on average, this was by 19.22 cm (+31.2%), whereas birches downregulated by 14.6 cm on average (−18.8%). The height growth of green alder, speckled alder, gray birch, and white birch were 89.3, 53.3, 72.3, and 67.9 cm, respectively (Figure 5A).

**Table 7.** Growth traits ANOVAs, including the source of variation, degrees of freedom (df), mean square values (MS), variance components (VC), *p*-values, and coefficient of determination ($R^2$). All *p*-values < 0.05 are in bold print.

| Source of Variation | df | Height Growth (cm) | | | Stem Dry Mass (g) | | |
|---|---|---|---|---|---|---|---|
| | | MS | VC (%) | *p*-Value | MS | VC (%) | *p*-Value |
| Block | 1 | 1189.83 | 0.7 | 0.064 | 906.38 | 5.6 | **<0.001** |
| Genus | 1 | 109.37 | 0.1 | 0.573 | 862.59 | 5.3 | **<0.001** |
| $CO_2$ | 1 | 400.17 | 0.2 | 0.282 | 4.58 | <0.1 | 0.738 |
| $CO_2$ × Genus | 1 | 21,197.31 | 11.7 | **<0.001** | 1531.62 | 9.4 | **<0.001** |
| Species (Genus) | 2 | 22,926.61 | 25.3 | **<0.001** | 404.54 | 5.0 | **<0.001** |
| $CO_2$ × Species (Genus) | 2 | 590.84 | 0.7 | 0.181 | 21.4 | 0.3 | 0.594 |
| Provenance (Spp.) | 14 | 540.03 | 4.2 | 0.087 | 23.3 | 2.0 | 0.888 |
| Provenance (Spp.) × $CO_2$ | 14 | 470.86 | 3.6 | 0.168 | 13.99 | 1.2 | 0.988 |
| Error | 282 | 343.97 | 53.6 | | 40.96 | 71.2 | |
| $R^2$ | | | 0.503 | | | 0.303 | |

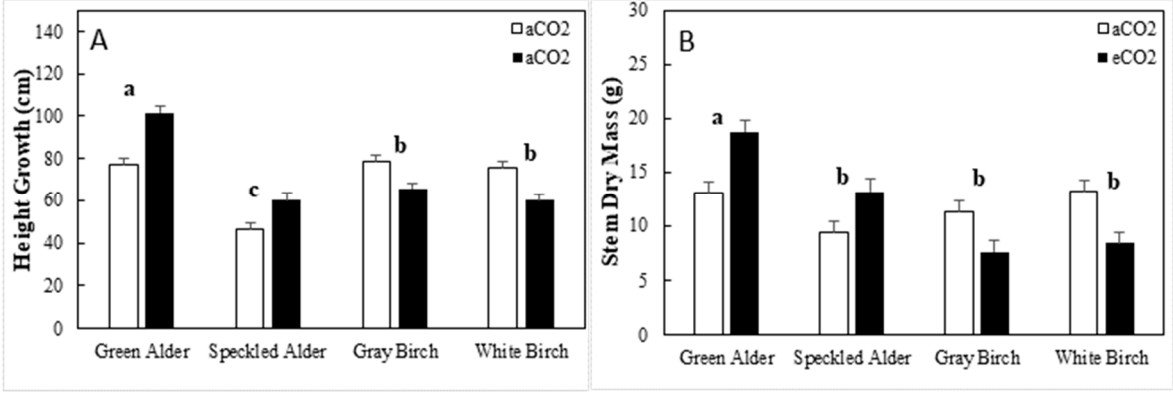

**Figure 5.** Growth traits (mean ± *SE*, *n* = 319) including, (**A**) height growth, and (**B**) stem dry mass by species and $CO_2$ treatment type ($aCO_2$ = ambient $CO_2$, $eCO_2$ = elevated $CO_2$). A post hoc Tukey's mean separation test was performed on species (*p* = 0.05) for each analysis where species was significant (Table 7). Species with the same letter are not significantly different.

For total stem dry mass, genus, species, and genus $\times$ $CO_2$ interaction were significant, accounting for 5.3, 5.0, and 9.4% of total variation, respectively (Table 7). The genus $\times$ $CO_2$ interaction was due to alders upregulating total stem dry mass by 4.8 g (+42.9%) and birches downregulating by 4.3 g ($-34.9$%). The average total stem dry mass of alders was 33% higher than that of birches, at 13.6 and 10.2 g, respectively. The total stem dry mass of green alder, speckled alder, gray birch, and white birch were 15.9, 11.3, 9.5, and 10.8 g, respectively (Figure 5B).

*3.4. Covariate Analysis*

Covariate analysis of $V_{cmax}$ in relation to N, testing the $CO_2$ effect, had no significant $CO_2 \times$ N interaction ($p = 0.746$). Further analysis showed a significant $CO_2$ effect ($p = 0.028$), and N response ($p = 0.003$), resulting in two separate parallel lines for which, in both, $V_{cmax}$ had a positive correlation with N. This correlation was greater under $eCO_2$ than under $aCO_2$ (Figure 6A). Covariate analysis of $A_{600}$ in relation to N, testing the $CO_2$ effect, had no significant $CO_2 \times$ N interaction ($p = 0.859$). Further analysis showed a significant $CO_2$ effect ($p = 0.009$) and N response ($p < 0.001$), resulting in two separate parallel lines for which, in both, $A_{600}$ had a positive correlation with N. This correlation was greater under $eCO_2$ than under $aCO_2$ (Figure 6B). Covariate analysis of $iWUE_{600}$ in relation to N, testing $CO_2$ effect, had no significant $CO_2 \times$ N interaction ($p = 0.188$) (Figure 6C). Further analysis showed a significant $CO_2$ effect ($p = 0.005$) and N response ($p = 0.001$), resulting in two separate parallel lines in which $iWUE_{600}$ had a positive correlation with N. This correlation was greater under $eCO_2$ than under $aCO_2$ (Figure 6C). Covariate analysis of $TCC$ in relation to N, testing the $CO_2$ effect, had a significant $CO_2 \times$ N interaction ($p = 0.049$) as well as significant $CO_2$ ($p = 0.047$) and N ($p = 0.013$), resulting in two lines with different slopes (Figure 6D). The slope under $eCO_2$ was less than that under $eCO_2$, indicating a lower increase in $TCC$ with increases in foliar N, compared to $aCO_2$.

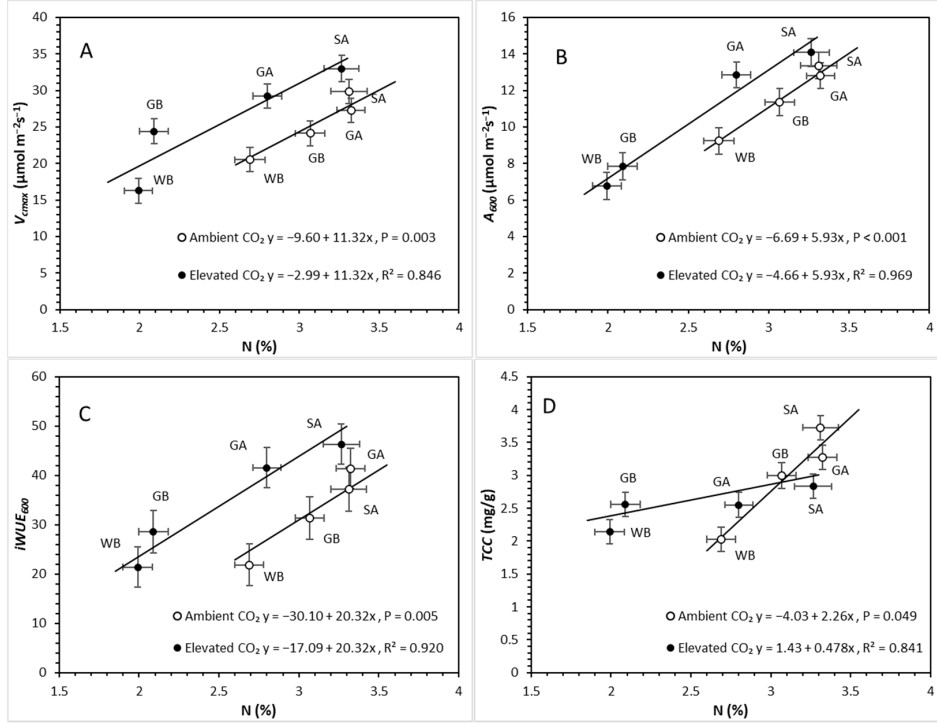

**Figure 6.** Relationships between (**A**) maximum rate of carboxylation ($V_{cmax}$) (mean $\pm$ SE), (**B**) assimilation at 600 ppm ($A_{600}$) (mean $\pm$ SE), (**C**) intrinsic water use efficiency ($iWUE_{600}$) (mean $\pm$ SE), and (**D**) total chlorophyll concentration ($TCC$) (mean $\pm$ SE) and foliar nitrogen concentration (N) (mean $\pm$ SE), testing $CO_2$ treatment as an independent effect across species (GA = green alder, SA = speckled alder, GB = gray birch, WB = white birch) under ambient and elevated $CO_2$ concentrations.

Covariate analysis of $V_{cmax}$ in relation to N, testing genus effect, had no significant genus $\times$ N interaction ($p = 0.777$) or genus effect ($p = 0.108$), but did have a significant N response ($p = 0.034$), resulting in a single line which indicated a positive correlation between $V_{cmax}$ and foliar N. An increase in N is associated with an increase in $V_{cmax}$ (Figure 7A). Covariate analysis of $A_{600}$ in relation to N, testing genus effect, had no significant genus $\times$ N interaction ($p = 0.130$), but did demonstrate a significant genus effect ($p = 0.033$) and a significant *N* response ($p = 0.007$), indicating that alders had a higher baseline assimilation rate, although the same positive correlation with N (Figure 7B). Covariate analysis of *iWUE* in relation to *N*, testing genus effect, had no significant genus $\times$ *N* interaction ($p = 0.689$) or significant *N* response ($p = 0.494$), but did have a significant genus effect, resulting in separate parallel lines with no slope ($p = 0.002$), indicating a lack of correlation with N, but genus differences in which alders had greater $iWUE_{600}$ (Figure 7C). Covariate analysis of *TCC* in relation to N, testing genus effect, had no significant genus $\times$ N interaction ($p = 0.368$) or genus effect ($p = 0.644$), but did have a significant N response ($p = 0.043$), resulting in a single response line, which indicated a positive increase in *TCC* as N increases (Figure 7D).

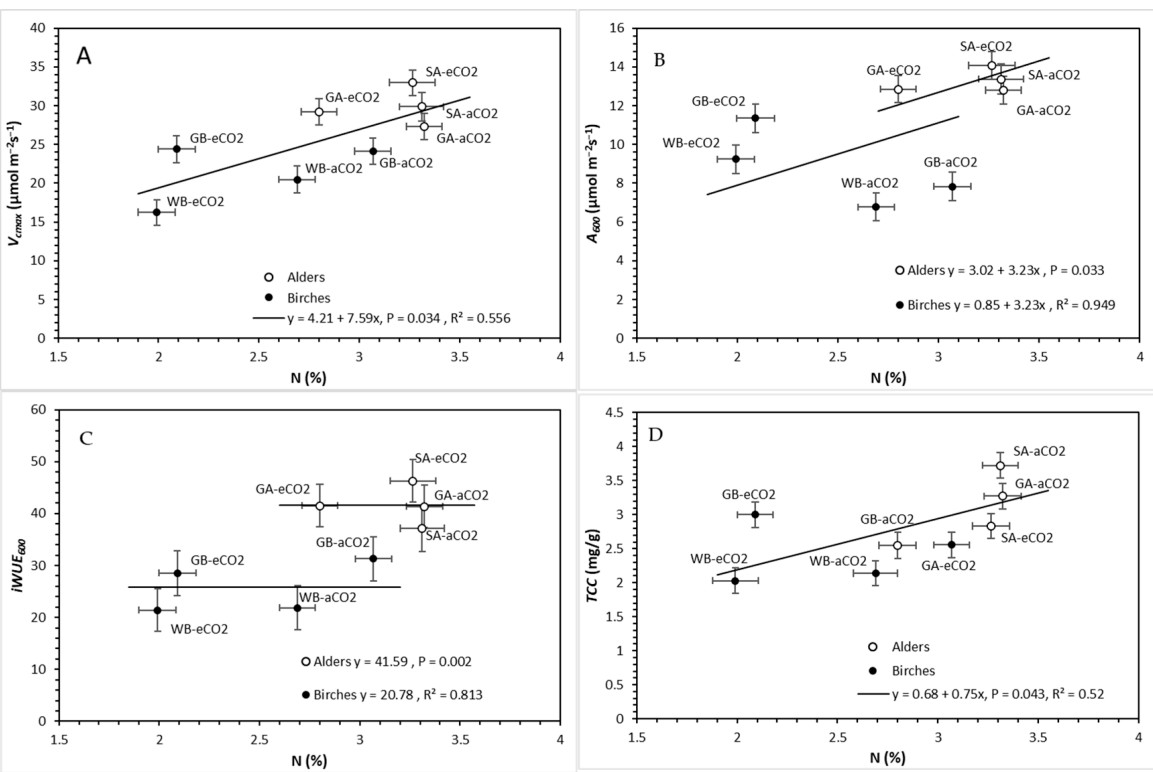

**Figure 7.** Relationships between (**A**) maximum rate of carboxylation ($V_{cmax}$) (mean $\pm$ *SE*), (**B**) assimilation at 600 ppm ($A_{600}$) (mean $\pm$ *SE*), (**C**) intrinsic water use efficiency ($iWUE_{600}$) (mean $\pm$ *SE*), and (**D**) total chlorophyll concentration (*TCC*) (mean $\pm$ *SE*) and foliar nitrogen concentration (N) (mean $\pm$ SE), testing genus as an independent effect under ambient and elevated $CO_2$ concentrations (a$CO_2$ = ambient $CO_2$, e$CO_2$ = elevated $CO_2$). Species in figures are abbreviated as GA = green alder, SA = speckled alder, GB = gray birch, and WB = white birch.

The simple linear regression of change in $A_{600}$ between a$CO_2$ and e$CO_2$ treatments ($\Delta A_{600}$) in relation to change in total stem dry mass between a $CO_2$ and e$CO_2$ treatments ($\Delta$Stem dry mass) (Figure 8) had a strong coefficient of determination, $R^2 = 0.842$, but was marginally significant ($p = 0.082,$). The alders clearly had greater positive change in assimilation and biomass stimulation compared to the birches; thus, the line was made into a dashed line to indicate marginal significance, which ranged between $p = 0.050$ and $p = 0.100$.

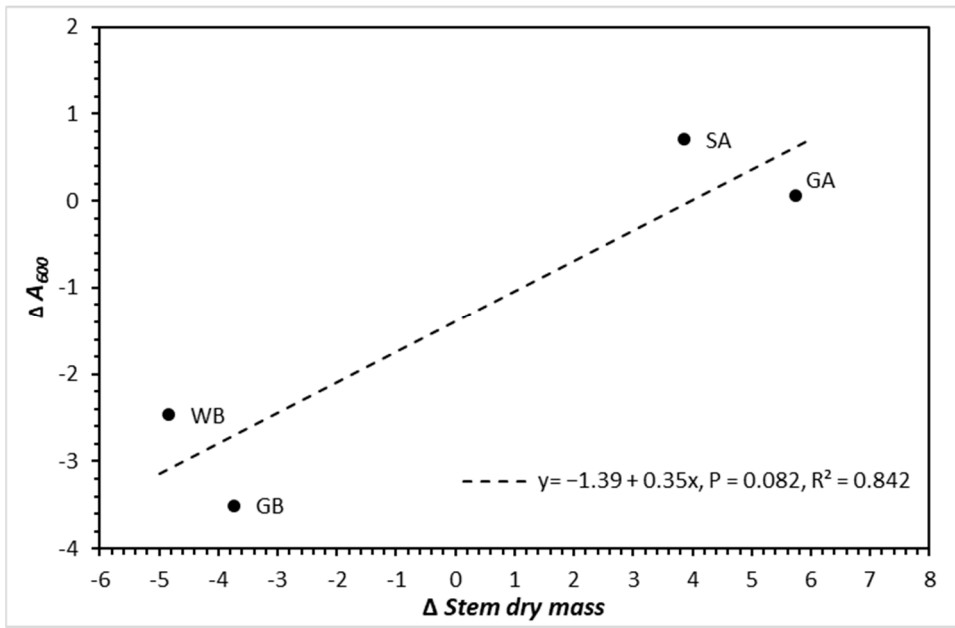

**Figure 8.** Delta assimilation (change in assimilation from ambient $CO_2$ to elevated $CO_2$) in relation to delta dry mass (change in dry mass from ambient $CO_2$ to elevated $CO_2$) by species (abbreviated as—GA = green alder, SA = speckled alder, GB = gray birch, and WB = white birch). Note that the line is dashed, as it was between $p = 0.050$ and $p = 0.100$.

## 4. Discussion

Consistent genus and species patterns for biochemical efficiency and some gas exchange (GE) responses to $eCO_2$ were found. Genus was most often the most influential factor, accounting for 23.9–41.2% of the total variation for maximum rate of carboxylation ($V_{cmax}$), maximum rate of electron transport ($J_{max}$), triose phosphate utilization (*TPU*), assimilation at 600 ppm ($A_{600}$), and intrinsic water use efficiency at 600 ppm ($iWUE_{600}$). Under $eCO_2$, these traits either remained the same or were upregulated in alders, whereas birches either downregulated or showed no change. Previous study [20] observed downregulation of $V_{cmax}$ and $J_{max}$ in white birch in response to $eCO_2$ and increased temperature, while [21] found increased stem dry mass growth in speckled alder and green alder in response to $eCO_2$, indicating greater photosynthetic efficiency. Stomatal conductance ($G_{wv600}$) is the only variable for which genus was not a significant factor and for which $CO_2$ had the largest effect, accounting for 15.4% of total variation, and downregulated for all species. Reduction in stomatal conductance is a common phenomenon in response to $eCO_2$, and thus it contributes to increased *iWUE* [11,12,22,23]. Alders had greater $iWUE_{600}$ than the birches, and this was driven by a greater $A_{600}$ and equal $G_{wv600}$ among genus and species.

Assimilation and assimilation-related traits have been found to downregulate over time in response to $eCO_2$ [3,24,25]. Greater atmospheric $CO_2$ can lead to declines in photosynthetic capacities, carboxylation efficiencies, electron transport, and chlorophyll concentrations [26,27], all of which we report here. The mechanism behind this can be a buildup of non-structural carbohydrates in the leaves, resulting from a lack of sink activity [28,29], and resulting in *TPU* limitation, which is a reduction in carbon exportation from the Calvin–Benson cycle [30]. We also see evidence of sink-driven photosynthesis by examining the changes in assimilation to changes in total stem dry mass from $aCO_2$ to $eCO_2$. This phenomenon has also been demonstrated by others [31–33].

The greater quantum yield of photosystem II ($\Phi_{PSII}$) and lower steady-state fluorescence (*RelFlu*) in alders indicated a greater fluorescence quenching due to photosynthesis compared to that in birches. The variance attributed to the genus × $CO_2$ interaction was 4.8%, and was driven by upregulation of $\Phi_{PSII}$ under $eCO_2$ for alders but downregulation of $\Phi_{PSII}$ in birches, similar to biochemical efficiency traits; however, there are few works in

the literature describing this trait or these genus $\times$ $CO_2$ interactions. The effects of $eCO_2$ on $\Phi_{PSII}$ in the current literature appear to be variable and dependent on the plants being used, with some publications finding decreases, and some finding instances of little to no difference [34–36]. Another possibility is that the downregulation of $\Phi_{PSII}$ is driven by nutrient limitation, primarily foliar N or phosphorus (P). Unfortunately, we did not measure leaf P concentrations. It has been suggested that some plants under $eCO_2$ have an increased sensitivity to P limitation, causing assimilation downregulation ($A_{dr}$), as well as downregulation of $\Phi_{PSII}$, quantum efficiency (Fv'/Fm'), and electron transport rate [34,37]. Additionally, a meta-analysis of P limitation on growth traits and gas exchange responses showed that P, when present in limited quantities, does inhibit *A* stimulation by $eCO_2$ and lowers *WUE* when in conjunction with $eCO_2$ [38], but we did not find this. Foliar N was measured and downregulated for all species under $eCO_2$ and had a significant genus $\times$ $CO_2$ interaction, which was a result of alders not downregulating N as much as did birches and was likely the main factor in birch assimilation downregulation. A N decrease is known to be a limiting factor, as it is a vital nutrient for the composition of the photosynthetic apparatus [39–41]. Relative fluorescence was driven only by genus and species effects in a direction contrary to the $\Phi_{PSII}$ and biochemical efficiencies. This result reflects the light energy that was not quenched by photosynthesis and was lost to fluorescence, and again, little exists in the literature on this trait under elevated $CO_2$, especially for the species used in our experiment.

Both a limited sink-effect and foliar N limitations were likely driving the greater downregulation in birches. Alders often grow multiple main shoots, as opposed to birches, which typically grow a single stem. This provides additional sink activity for alders, a capacity which aligns with our total stem dry mass results. The positive linear regression of change in $A_{600}$ to change in total stem dry mass supports the sink-driven photosynthetic regulation. Maximum rate of carboxylation also has a positive relationship to foliar N, and genus was not a significant factor; however, there was a strong genera segregation along the same response line driven by alders having greater N concentration and $V_{cmax}$. Genus was significant in the $A_{600}$ analysis, indicating greater overall $A_{600}$ and N values for alders compared to birches. Genus and species differences can often occur for assimilation efficiencies and growth traits, especially when examining fertilized and non-fertilized treatment effects [3,42,43]. Both alder species used in this study are actinorhizal and form symbiotic relationships with *Frankia alni* bacteria [6], which allows the species to fix atmospheric N through the formation of root nodules. Other studies indicate that N fertilization given to plants under $eCO_2$ can restore biochemical efficiencies [3,44]. As birches do not form this symbiosis, foliar N concentrations were lower, and, thus, so were the assimilation and biochemical efficiencies. This linear relationship between $V_{cmax}$ and foliar N concentration and testing $CO_2$ treatments has been found before in *Pinus radiata* [45], sitka spruce [46], and soybeans [47]. The biochemical efficiency responses to $eCO_2$ are consistent with height growth and total stem dry mass; alders showed increased height growth and total stem dry mass in response to $eCO_2$, whereas these values for birches were slightly reduced.

Assimilation at 600 ppm had a positive relationship to foliar N under both $aCO_2$ and $eCO_2$, which drove the positive $iWUE_{600}$ to foliar N relationship, along with the $G_{wv600}$ decrease in response to $eCO_2$. A study involving red alder (*Alnus rubra* Bong.) found a similar effect when analyzing *iWUE* (*A/g*), wherein increased *iWUE* was a product of increased *A* for a given stomatal conductance [48]. This would explain why $iWUE_{600}$ values were lower for birches, given that N, and thus $A_{600}$, had greater downregulation under $eCO_2$. Ref. [49] found that low N concentrations lowered dry matter in *Canna edulis* Ker-Gawler, and subsequently lowered *iWUE*. This was proposed to be a function of lower assimilation. Does this mean that *iWUE* is always driven by *A* rather than $G_{wv}$ under $eCO_2$? Not necessarily, and a search of the current literature finds inconsistent responses. Downregulation of $G_{wv}$ in response to $eCO_2$ appears to be common [11,22], but other studies have found little to no change [3,50,51].

Total chlorophyll content and *CAR* decreased for alders and gray birch and were not consistent among species. Thus, *TCC* and *CAR* do not appear to be main factors driving alder upregulation. As both alder species used in this study are actinorhizal, it appears that foliar N is likely not limiting alder photosynthetic capacity, as previously discussed. Our foliar N analysis shows that downregulation of *TCC* and *CAR* appears to be related to N downregulation, but not necessarily completely in balance. This relationship is confirmed through the covariate analysis of *TCC* and N, with genus as a covariate (Figure 7D). Genus itself was not significant, but instead, genus was segregated on the same covariate line. Under eCO$_2$, the decreases in foliar N for green alder and speckled alder were 15.5% and 7.9%, respectively, whereas the decreases were 29.9% and 26.6% for gray birch and white birch. Foliar N had a greater effect on *TCC* under aCO$_2$ than under eCO$_2$, which is indicated by a more gradual slope between *TCC* and N (Figure 6D). Lower *TCC* under eCO$_2$ is often seen in the literature [25,46,52–54], and usually found alongside a decrease in foliar N, which is interpreted as a reallocation of N away from the photosynthetic apparatus [25,54]. The greater foliar N and the symbiotic relationship with *Frankia alni* bacteria explains why alders exhibited less N downregulation than did birches under eCO$_2$, although this does not explain why *TCC* and CAR did not downregulate as much for birches, as their foliar N downregulated more.

For first-year height growth, genus only accounted for 0.1% of the total variation, which means that, despite the many physiological differences we found between genera, height growth between genera were very similar. Species accounted for half of the defined sources of variation: 25%. This means that the height growth variation between species within genus was highly variable. Provenances (seed sources) accounted for 4.2% of the total variation, and this was marginally significant ($p = 0.087$). In contrast, for total stem dry mass, genus accounted for 5.3% of the total variation, and this was highly significant; this is due to the multiple-stem trait of alders, compared to birches, which in most cases only have one stem. Species within a genus for total stem dry mass accounted for 5 times less variation (5%) than height growth, and provenance accounted for less than 2% of the total variation for total stem dry mass. It should be noted that most of the seed sources were from various parts of the New Brunswick province, with a few originating from the neighboring provinces of Nova Scotia and Prince Edward Island. It is important to consider that these results are only for first-year growth, and warrant some caution. Why does genus account for between 23–42% of total variation for gas-exchange-related traits and, for chlorophyll content, between 10–13% of total variation, but height growth and total stem dry mass genus variation only account for 0.1–5% of the variation? The answer is in the second-year dry mass results. For coppice regrowth, genus accounted for 34% of height growth variation and an astounding 74% of total stem dry mass variation in the second year [55]. It appears that the physiological traits are working towards the later, second-year dry mass trait results. In addition, the results found in first- and second-year growth on barren restoration sites [56] found small differences among genus and species in the first year, but up to approximately 13 times greater dry mass for alders, as compared to birches, in the second year.

## 5. Conclusions

Alders and birches are early successional deciduous species from the same phylogenetic family, Betulaceae. Alders had greater biochemical efficiencies, assimilation, intrinsic water use efficiency, photochemical yield, chlorophyll pigments, and foliar N than birches. Alders had a greater positive response to eCO$_2$ than did birches for these traits. This was reflected in total stem dry mass results in response to eCO$_2$. Alders are actinorhizal and form symbioses with *Frankia alni* bacteria, which allows the genera to fix atmospheric N through the formation of root nodules in exchange for photosynthates. This relationship allowed alders to take advantage of eCO$_2$, mitigate assimilation downregulation, and sequester more carbon, unlike the birches. Covariate analysis examining carbon assimilation traits in relation to foliar N showed clear positive physiological functional responses to foliar

N. Both species in each genus behaved similarly. With atmospheric $CO_2$ continuing to increase, restoration planting with alders secures greater carbon sequestration and would accelerate the reclamation of impoverished sites.

**Author Contributions:** J.E.M. designed and co-analyzed the experiment and co-authored the manuscript. A.B. managed the experiment, co-analyzed the data, and co-authored the manuscript. All authors have read and agreed to the published version of the manuscript.

**Funding:** This research received Canadian Department of National Defense (DND) Environmental Services Branch (ESB) funding and was supported by the Department of Natural Resources Canada.

**Data Availability Statement:** Contact corresponding author regarding data.

**Acknowledgments:** We gratefully acknowledge the useful edits and comments received from John Kershaw, Alex Mosseler, Myriam Barbeau, and Jasen Golding. We are also grateful for the support from Noah Pond, Deanna McCullum, and Meagan Betts from DND ESB. In addition, we are grateful to the following for the technical help in the establishment and management of the experiment: Dominic Galea, Shawn Palmer, John Malcom, Will Bradley, Megan Hall, and Josh Kilburn.

**Conflicts of Interest:** The authors declare no conflicts of interest.

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
