# Peer review of "Effects of CO2 Treatments on Functional Carbon Efficiencies and Growth of Forest Tree Seedlings: A Study of Four Early-Successional Deciduous Species"

_forests, doi:10.3390/f15010193_

Round 1

Reviewer 1 Report

Comments and Suggestions for Authors

Comments:

The title of the manuscript is very interesting to readers.

Abstract

The abstract of the manuscript needs to add a few lines of introduction, conclusion, and practical application of the present study.

Kindly delete the highlighted words from abstracts; they are already written in the introduction section.

Introduction

The use of the abbreviation of the common name of species (Speckled alder) as SA, is not scientific language. So kindly delete it from the entire manuscript.

Write the number of seeds sown and the number of seedlings placed in bags at -4 to -5°C in a freezer.

Please incorporate some photographs of the study species in the manuscript.

Materials and methods

The materials and methods section of the manuscript needs to improve as per the suggestions.

Results

The results section of the manuscript is written very nicely.

Discussion

The discussion of the manuscript must be improved by adding some recent relevant references.

Conclusion

The conclusion of the study must be improved.

The scientific name of the study species must be written throughout the results, figures, and discussion section.

I hope my submitted comments help in improving the manuscript.

Reviewer 2 Report

Comments and Suggestions for Authors

- Please revise the abstract and reduce the length. Focus on important content. 

- Line 55: Avoid starting sentences with abbreviations. Check throughout MS.

- Lines 97-131: check the font and font size. Check throughout MS and keep consistency. 

- Figure 1: Please provide the description of letters for statistical comparison in the figure description. Its confusing. Are they indicating statistical comparison between aCo2 and eCo2, or for aCo2 among different species or for eCo2 among different species. What is the objective here? to show how species are different under aCo2 and eCo2 or how each specie is different under aCo2 and eCo2 or.......?

- Figure 2: Same question above as Figure 1. Provide description and no letters at 2B?

- Figures 3-5: Provide description of letters

- Figure 6,7: What are these error bars and what are they indicating  (SD, SE...?). Provide description in caption.

- Check the references format. Italicize the scientific names. 

Comments on the Quality of English Language

Moderate editing of English language required. 

Reviewer 3 Report

Comments and Suggestions for Authors

General comments

The manuscript examines in a field experiment the seedlings of four early successional tree species (Alnus viridis, A. incana, Betula populifolia, B. papyrifera) that are important in regional nature conservation practice. The examined parameters are focused on the carbon exchange of plants, including cellular and individual functional responses, with three A/Ci, two fluorescence, and three gas exchange parameters. The data collection focuses on the early phase of seedling development, according to the methodological description, with 1-year-old tree seedlings propagated from seed and pre-treated by trimming (1 year of preparation + 1 year of measurement). The plants were grown under two types of carbon dioxide treatment (normal and elevated levels) in a random block design in open-air chambers. The evaluation of the experiment mainly focuses on the parameter differences due to the carbon dioxide treatments, but the provenance, the tree species and genera, the functional parameters and their interactions are also important factors. The authors apply a statistical evaluation model by ANOVA, focus on the actually effective factors, and also present non-effective (p>0.050) cases. Based on these, it can be concluded that the experiment was systematically planned in detail and carefully implemented. The conclusions drawn are correct (except the questionable cases, which I will detail later), but in some cases, they are too detailed, they do not focus on the main phenomenon, and therefore it is difficult to review. On the contrary, the description of the methods and the presentation of the results contain several major and minor inconsistencies and shortcomings, which must be corrected for the sake of credibility, consistency, and easier understanding.

The title of the manuscript does not fully cover the applied protocol and does not focus on the essence of the experiments (i.e. general and specific functional carbon exchange responses of early successional tree species seedlings). The height and biomass yield of the plants are not simple morphological characteristics, please take this into account. Therefore, I recommend rewording the content and shortening the title, as it will become clear later which tree species and parameters were examined. The entire manuscript is subordinated to abbreviations, so identification and comparisons of parameters are very difficult to follow, and in many cases, the essence is lost. I recommend that you do not use abbreviations at all in chapters that are expected to be comprehensible (Abstract, Introduction, Conclusion, titles of sub-chapters), rather write the general designation/full name of the parameters/species, so that not only professionals dealing with ecophysiology in the narrow sense can understand it. The explanations of the figures and tables are incomplete everywhere, they do not include the overall introduction (this must be replaced anyway), but they only list the listed parameters again, this is a repeated statement that must be eliminated. Most of the information entered inside the sub-figures should be removed (most of them are included in the previous table), and the symbol key should be transferred to the description of the figure. It is also not always clear in which cases species are considered separately (i.e. 4 species,  df=3) and genera (i.e. 2 genera, df=1) as sources of variations. If Species (Genus) and Provenance (Spp.) are not interaction factors, they will clearly be more important. If they are, then mark them accordingly and highlight those cases in the covariation chapter where they are really important. There is also a great deal of inconsistency in specifying/listing the carbon dioxide levels used: in the general parts, 800 ppm, in the methodological part, a series, e.g. It is listed in 11 steps between 50-2200 (presumably the measurement points used for the A/Ci curve), and in the results, we find statements about 600 ppm without justification - why not at 800 ppm? Could it be that the levels of growth and measurement were different? Please take a clear position on which carbon dioxide treatments were used consistently and report the results related to them, any partial results relating to lower or higher carbon dioxide levels should be omitted. It is necessary to specify the water use parameter in the entire manuscript with a relevant reference (i.e. iWUE). Authors attach great importance to covariates, even though they provide mixed results, and were applied for leaf nitrogen and dry mass change to assimilation change. It is worth considering again which of the obtained relationships can be generalized to the case of functional convergence beyond the study species.

The manuscript needs to be re-examined and corrected according to the aspects described in general and detailed comments. Please highlight the changed/modified parts in color in the next version of the manuscript.

Detailed suggestions

Lines 2-4   I suggest rewriting the title based on the essence of the experiments and objects

Lines 9-35   In the abstract, a short methodological description is missing, it makes up for it; please specify the level of CO2 treatments used; please eliminate abbreviations and emphasize the points resulting in a more consistent presentation i.e. the similarities and differences between species and genera in the parameter groups, with a specific focus on the most important cross-effects/interactions.

Lines 11-12   indicate the subspecies rank using the taxonomic standard - note that it would be better to list the English names here, later in the introduction they must be listed using their full scientific names

Line 36   specify the common names according to the number of species (e.g. alders); 'differential' is not necessary for assimilation feedback (keyword is better in general form)

Lines 74-85   in this section, the 'quantification of genetic variations' is misunderstood, because there is no real information about the genetic difference or its type/extent, I recommend omitting it or rewording it

Line 87   clarify the term 'treatment delivery' (would it be a treatment protocol?)

Lines 93-95  specify the time/placement of the trimming along the seedling development

Line 97   please explain the meaning of the abbreviation AFC

Lines 102-105   the planting medium used was soil (natural and original) or artificial composition (mixed from components), please clarify

Line 110-111   enter the name of the method according to the reference [17]

Lines 119-121   specify the levels of the applied CO2 treatments in line with the rest of the manuscript and enter the standard deviation ranges instead of approx., growth, and measurement levels were different - iIf so, please specify this as well

Line 126   check the element abbreviations in Plant-Prod “Forestry Starter” 11:41:8, 287g/25L, +250ml of MgNiFeCal fertilizer, if the last one is calcium, correct it

Lines 143-145   in the title of the table, give a general description instead of the repeated list of parameters; enter the meaning of the abbreviations or write the full name (the latter would be consistent); mark the subspecies; provide more detailed information on the origin of Alnus incana (instead of bulk)

Line 147   if it is leaf chlorophyll and leaf nitrogen, please mark/specify

Line 149   'LI-COR' appears twice, delete one

Lines 150-155   A/Ci must be indicated uniformly (here and in throughout the manuscript)

Lines 156-157   Instead of PAR, I recommend using PPFD (a specific term/ parameter in ecophysiology)

Lines 159-160   please clarify/refine and reconcile CO2 levels (here and throughout the manuscript)

Lines 170-171   CO2 level at 600 ppm is not a 'common' level, or if so, what does 400 ppm mean - clarify and reconcile with the rest of the manuscript

Lines 172-173   specify the type of water use efficiency (i.e. iWUE)

Line 178   I recommend changing the abbreviation (N) to a full term in the chapter title (see Chapter 3.3)

Line 210   how/with what was the height of the main stem of the plants measured, please specify

Line 213   enter accuracy and technical specifications for the precision scale

Line 217  enter the type of ANOVA used, if there were several, then all of them

Lines 240-241   enter the name of the method according to the reference [19]

Lines 253-255   specify which confidence level you accepted in the evaluation of the experiments

Lines 259-265   the properties of the soil/planting medium were not an experimental factor (I could not find a CO2×soil experiment description), only a background factor - their description does not belong to the results, I recommend moving it to the methodology chapter; note that the sand:silt:clay ratio is reported three times in the manuscript, eliminate two of them; the other parameters can remain in a simple table or as a list in the text

Line 266   I recommend changing the abbreviation (GE) to a full term in the chapter title

Lines 285-288   in Table 3, add a general title to the table (what is in it), do not list the listed parameters again; A600 is not stated, instead Jmax; in the table, arrange the sources of variation in some logical order, e.g. simple and interaction factors, within them in order of importance/effectiveness, etc...; clarify the content of 'Species (Genus)' because it is not clear what the difference is compared to 'Genus'; if it is about the four species, put it in the group of simple factors and df=3

Lines 289-294   in Figure 1, give a general description of what is shown in the figure (e.g. results of A/Ci curves), do not list the presented parameters again; eliminate the data in the subfigures, because they have already been published once in Table 1 (double presentation); enter the meaning of the CO2 levels in the description of the figure (as a single code); cancel the label on the x-axis, because it is clear that the species are sorted

Lines 308-312   in Table 4, see suggestions for Table 3

Lines 313-318   in Figure 2, see suggestions for Figure 1

Lines 329-333   in Table 5, see suggestions for Tables 3 and 4

Lines 334-338   in Figure 3, see suggestions for Figures 1 and 2

Lines 362-366   in Table 6, see suggestions for Tables 3,4 and 5

Lines 334-338   in Figure 4, see suggestions for Figures 1,2 and 3

Line 372   the two parameters refer to similar plant parts (main stem and all stems), I recommend that this be expressed in the title as well, e.g. stem height and dry mass and also in other places in the manuscript (pl. Table 7, etc.)

Lines 384-387   in Table 7, see suggestions for Tables 3,4,5 and 6; what does ANOVAs refer to in line 384?

Lines 388-392   in Figure 5, see suggestions for Figures 1,2,3 and 4

Lines 430-436 … in Figure 6, ambient and elevated CO2 are double marked (letters A-E and white-black symbols), delete one; match the markings with the rest of the manuscript (ACO2 versus aCO2)

Lines 434-442    in Figure 7, see suggestions for Figure 6, and the colors of the symbols refer to two test species each, indicate this in the names as well (plural)

Lines 522-524    how would you evaluate the downregulation of Gwv in response to eCO2 based on your current study?

Lines 525-541    what parameter is CAR, I didn't read about it anywhere else in the manuscript, please explain/justify

Line 547   p=0.087 is not significant (meaning that p>0.050), with that in mind, rethink this paragraph

Lines 566-582   in this chapter, remove abbreviations (use only text explanations), and rethink the scientific (e.g. functional convergence) and practical conclusions (e.g. assimilation upregulation versus downregulation, species specifics, genera specifics, most effective and important covariate parameters) by pointing out the most relevant results

23-12-2023

Comments on the Quality of English Language

In terms of English language application, I noticed a lot of errors/mistakes throughout the manuscript - i.e. lack of definite article,  improper matching of singular and plural, the spelling of connected expressions, e.g. gas exchange-related (instead of gas exchange-related), one year versus first-year, etc. Please review the manuscript carefully again from this point of view.

Round 2

Reviewer 3 Report

Comments and Suggestions for Authors

Thanks for your responses and corrections, it made the manuscript much more consistent and clear. I would suggest a few more clarifications as follows.

The title is still not specific enough, because it is not simply a matter of deciduous species, but of forest tree seedlings, please clarify accordingly

For the methodological part of the abstract, write an extended sentence about the statistical analysis (what type/for what aim was it carried out) and insert it in line 16

I still think that it is not the genetic difference between species and genera that are evaluated, but the functional responses/characteristics based on it/expressed as a result of it - please reconsider this for the sake of the research aim

Explaining the type of statistical analysis also made the results much more understandable. It would be even better if a more well-known name appeared in the methodology section - i.e. General Linear Model (GLM) with effect size indicator (R2) - if this is the case...

Lines 162-163   the listed values are not consistent with the 11 measurement levels mentioned in the text, please clarify

10-01-2024
